# Learning Representations for Reinforcement Learning with Hierarchical Forward Models

## Abstract

Learning control from pixels is difficult for reinforcement learning (RL) agents because representation learning and policy learning are intertwined. Previous approaches remedy this issue with auxiliary representation learning tasks, but they either do not consider the temporal aspect of the problem or only consider single-step transitions, which may miss relevant information if important environmental changes take many steps to manifest. We propose Hierarchical $k$-Step Latent (HKSL), an auxiliary task that learns representations via a hierarchy of forward models that operate at varying magnitudes of step skipping while also learning to communicate between levels in the hierarchy. We evaluate HKSL in a suite of 30 robotic control tasks with and without distractors and a task of our creation. We find that HKSL either converges to higher or optimal episodic returns more quickly than several alternative representation learning approaches. Furthermore, we find that HKSL's representations capture task-relevant details accurately across timescales (even in the presence of distractors) and that communication channels between hierarchy levels organize information based on both sides of the communication process, both of which improve sample efficiency.

## 1 Introduction

Recently, reinforcement learning (RL) has had significant empirical success in the robotics domain (Kalashnikov et al., 2018; 2021; Lu et al., 2021; Chebotar et al., 2021). However, previous methods often require a dataset of hundreds of thousands or millions of agent-environment interactions to achieve their performance. This level of data collection may not be feasible for the average industry group. Therefore, RL's widespread real-world adoption requires agents to learn a satisfactory control policy in the smallest number of agent-environment interactions possible.

Pixel-based state spaces increase the sample efficiency challenge because the RL algorithm is required to learn a useful representation and a control policy simultaneously. A recent thread of research has focused on developing auxiliary learning tasks to address this dual-objective learning problem. These approaches aim to learn a compressed representation of the high-dimensional state space upon which agents learn control. Several task types have been proposed such as image reconstruction (Yarats et al., 2020; Jaderberg et al., 2017), contrastive objectives (Laskin et al., 2020a; Stooke et al., 2021), image augmentation (Yarats et al., 2021; Laskin et al., 2020b), and forward models (Lee et al., 2020a; Zhang et al., 2021; Gelada et al., 2019; Hafner et al., 2020; 2019).

Forward models are a natural fit for RL because they exploit the temporal axis by generating representations of the state space that capture information relevant to the environment's transition dynamics. However, previous approaches learn representations by predicting single-step transitions, which may not capture relevant information efficiently if important environmental changes take many steps to manifest. For example, if we wish to train a soccer-playing agent to score a goal, the pertinent portions of an episode occur at the beginning, when the agent applies a force and direction, and at the end when the agent sees how close the ball came to the goal. Using multi-step transitions in this situation could lead to more efficient learning, as we would focus more on the long-term consequences and less on the large portion of the trajectory where the ball is rolling.

In this paper, we introduce *Hierarchical $k$-Step Latent* (HKSL)[1], an auxiliary task for RL agents that explicitly captures information in the environment at varying levels of temporal coarseness. HKSL accomplishes this by leveraging a hierarchical latent forward model where each level in the hierarchy predicts transitions with a varying number of steps skipped. Levels that skip more steps should capture a coarser understanding of the environment by focusing on changes that take more steps to manifest and vice versa for levels that skip fewer steps. HKSL also learns to share information between levels via a communication module that passes information from higher to lower levels. As a result, HKSL learns a set of representations that give the downstream RL algorithm information on both short- and long-term changes in the environment.

We evaluate HKSL and various baselines in a suite of 30 DMControl tasks (Tassa et al., 2018; Stone et al., 2021) that contains environments without and with distractors of varying types and intensities. Also, we evaluate our algorithms in "Falling Pixels", a task of our creation that requires agents to track objects that move at varying speeds. The goal in our study is to learn a well-performing control policy in the smallest number of agent-environment interactions as possible. We test our algorithms with and without distractors because real-world RL-controlled robots need to work well in controlled settings (e.g., a laboratory) and uncontrolled settings (e.g., a public street). Also, distractors may change at speeds independently from task-relevant information, thereby increasing the challenge of relating agent actions to changes in pixels. Therefore, real-world RL deployments should explicitly learn representations that tie agent actions to long- and short-term changes in the environment.

In our DMControl experiments, HKSL reaches an interquartile mean of evaluation returns that is 29% higher than DrQ (Yarats et al., 2021), 74% higher than CURL (Laskin et al., 2020a), 24% higher than PI-SAC (Lee et al., 2020b), and 359% higher than DBC (Zhang et al., 2021). Also, our experiments in Falling Pixels show that HKSL converges to an interquartile mean of evaluation returns that is 24% higher than DrQ, 35% higher than CURL, 31% higher than PI-SAC, and 44% higher than DBC. We analyze HKSL's hierarchical model and find that its representations more accurately capture task-relevant details earlier on in training than our baselines. Additionally, we find that HKSL's communication manager considers both sides of the communication process, thereby giving forward models information that better contextualizes their learning process. Finally, we provide data from all training runs for all benchmarked methods.

## 2 BACKGROUND

We study an RL formulation wherein an agent learns a control policy within a partially observable Markov decision process (POMDP) (Bellman, 1957; Kaelbling et al., 1998), defined by the tuple $(\mathcal{S}, \mathcal{O}, \mathcal{A}, P^s, P^o, \mathcal{R}, \gamma)$. $\mathcal{S}$ is the ground-truth state space, $\mathcal{O}$ is a pixel-based observation space, $\mathcal{A}$ is the action space, $P^s : \mathcal{S} \times \mathcal{A} \times \mathcal{S} \to [0, 1]$ is the state transition probability function, $P^o : \mathcal{S} \times \mathcal{A} \times \mathcal{O} \to [0, 1]$ is the observation probability function, $\mathcal{R} : \mathcal{S} \times \mathcal{A} \to \mathbb{R}$ is the reward function that maps states and actions to a scalar signal, and $\gamma \in [0, 1)$ is a discount factor. The agent does not directly observe the state $s_t \in \mathcal{S}$ at step $t$, but instead receives an observation $o_t \in \mathcal{O}$ which we specify as a stack of the last three images. At each step $t$, the agent samples an action $a_t \in \mathcal{A}$ with probability given by its control policy which is conditioned on the observation at time $t$, $\pi(a_t|o_t)$. Given the action, the agent receives a reward $r_t = \mathcal{R}(s_t, a_t)$, the POMDP transitions into a next state $s_{t+1} \in \mathcal{S}$ with probability $P^s(s_t, a_t, s_{t+1})$, and the next observation (stack of pixels) $o_{t+1} \in \mathcal{O}$ is sampled with probability $P^o(s_{t+1}, a_t, o_{t+1})$. Within this POMDP, the agent must learn a control policy that maximizes the sum of discounted returns over the time horizon $T$ of the POMDP's episode: $\arg\max_\pi \mathbb{E}_{a \sim \pi}[\sum_{t=1}^{T} \gamma^t r_t]$.

## 3 RELATED WORK

**Representation learning in RL.** Some research has pinpointed the development of representation learning methods that can aid policy learning for RL agents. In model-free RL, using representation learning objectives as auxiliary tasks has been explored in ways such as contrastive objectives (Laskin et al., 2020a; Stooke et al., 2021), image augmentation (Yarats et al., 2021; Laskin et al., 2020b), image reconstruction (Yarats et al., 2020), information theoretic objectives (Lee et al.,

---

[1] https://anonymous.4open.science/r/hksl-0D60/README.md

2020b), and inverse models (Burda et al., 2019; Pathak et al., 2017). HKSL fits within the auxiliary task literature but does not use contrastive objectives, image reconstruction, information theoretic objectives, nor inverse models.

**Forward models and hierarchical models.** Forward models for model-free RL approaches learn representations that capture the environment's transition dynamics via a next-step prediction objective. Some methods learn stochastic models that are aided with image reconstruction (Lee et al., 2020a) or reward-prediction objectives (Gelada et al., 2019). Other methods combine forward models with rewa rd prediction and bisimulation metrics (Zhang et al., 2021) or momentum regression targets (Schwarzer et al., 2021). Outside of the purpose of representation learning, forward models are used extensively in model-based RL approaches to learn control policies via planning procedures (Hafner et al., 2020; 2019; Ha & Schmidhuber, 2018; Zhang et al., 2019).

Stacking several forward models on top of one another forms the levels of a hierarchical model. This type of model has been studied in the context of multiscale temporal inference (Schmidhuber, 1991), variational inference Chung et al. (2017), and pixel-prediction objectives (Kim et al., 2019; Saxena et al., 2021). Additionally, hierarchical models have been used for speech synthesis (Kenter et al., 2019), learning graph embeddings (Chen et al., 2018), and decomposing MDPs (Steccanella et al., 2021). Sequence prediction literature has explored the use of hierarchical models via manually-defined connections between levels (Saxena et al., 2021; Koutnik et al., 2014) and using levels with uniform time-step skipping (Kumar et al., 2020; Castrejon et al., 2019).

Unlike the aforementioned forward model approaches, HKSL combines a set of forward models that step in the latent space with independent step sizes without additional prediction objectives. Also, HKSL contains a connection between forward models that learns what to share by using the context from the entire rollout from higher levels and the current timestep of lower levels, which leads to faster learning.

## 4  HIERARCHICAL $k$-STEP LATENT

HKSL's hierarchical model is composed of forward models that take steps in the latent space at varying levels of *temporal coarseness*. We define temporal coarseness as the degree to which a level's forward model skips environment steps. For example, if a forward model predicts the latent representation of a state five steps into the future, it is considered more coarse than a forward model that predicts only one step forward. Coarser levels should learn to attend to information in the environment that takes many steps to manifest in response to an agent's action. In contrast, finer levels should learn to attend to environmental properties that immediately respond to agent actions. This is because coarser levels need to make fewer predictions to reach steps further into the future than finer levels.

At each learning step, a batch of $B$ trajectories of length $k$ are sampled from the replay memory $\tau = \{(o_t, a_t, \ldots, a_{t+k-1}, o_{t+k})_i\}_{i=1}^{B}$. The initial observation of each trajectory $o_t$ is uniformly randomly sampled on a per-episode basis $t \sim U(1, T - k)^2$. In the following, we will denote the first and last timestep of the batch with $t = 1$ and $t = k$, respectively.

**HKSL's components.** See Figure 1 for a visual depiction of the HKSL architecture. HKSL's hierarchical model is composed of $h$ levels. Each level $l$ has a forward model $f^l$, a nonlinear projection module $w^l$, an online image encoder $e_o^l$, and a momentum image encoder $e_m^l$ that is updated as an exponential moving average of the online encoder (e.g., (He et al., 2020)). Between consecutive levels there is a communication manager $c^{l,l-1}$ to pass information from one level $l$ to the level below it $l - 1$. The number of steps skipped by a given level $n^l$ is independent of the coarseness of other levels in the hierarchy.

**Forward models.** HKSL's forward models are a modified version of the common GRU recurrent cell (Cho et al., 2014) that allows for multiple data inputs at each step. See Appendix C.3 for a detailed mathematical description. At step $t = 1$, the forward models take the representation produced by the level's encoder $z_1^l = e_o^l(o_1)$ along with a concatenation of $n^l$ action vectors $\bar{a}_1 = [a_1|...|a_{n^l}]$ to predict the latent representation of a future state $z_{1+n^l}^l = f^l(z_1^l, \bar{a}_1)$. For any

---

²Ending the range of numbers on $T - k$ guarantees that trajectories do not overlap episodes.

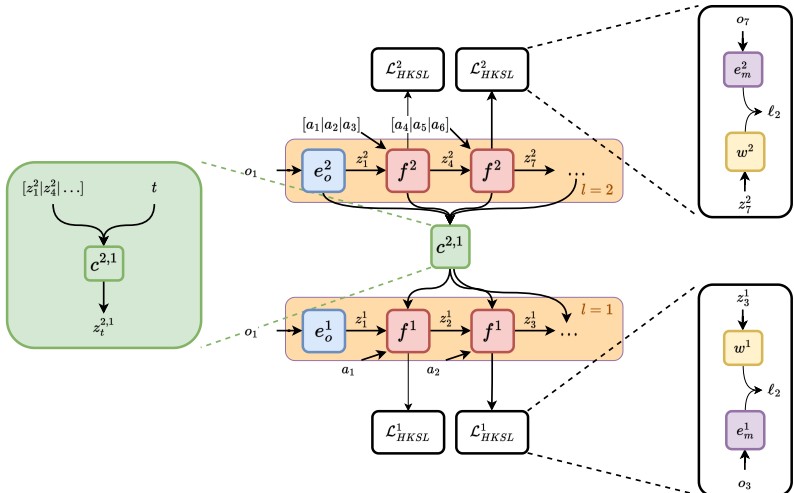

Figure 1: Depiction of HKSL architecture with an "unrolled" two-level hierarchical model where the first level moves at one step $n^1 = 1$ and the second level moves at three steps $n^2 = 3$. First, the online encoders (blue) encode the initial observation $o_1$ of the sampled trajectory. Next, the forward models (red) predict the latent representations of the following observations, with level 1 predicting single steps ahead conditioned on the level's previous representation and applied action. The forward model of the second level predicts three steps ahead and receives the previous representation and concatenation of the three applied actions. The communication manager (green) forwards information from the representations of the coarser second level to each forward model step of the first level as additional inputs. All models are trained end-to-end with a normalized $\ell_2$ loss of the difference between the projected representations of each level and timestep and the target representations of observations at the predicted timesteps. Target representations are obtained using momentum encoders (purple) and projections are done by the projection model (orange) of the given level.

following timestep $t > 1$, the forward models take the predicted latent representation $z_t^l$ as input instead of the encoder representation.

**Communication managers.** Communication managers $c^{l,l-1}$ pass information from coarser to finer levels in the hierarchy ($l \to l-1$) while also allowing gradients to flow from finer to coarser levels ($l-1 \to l$). A communication manager $c^{l,l-1}$ takes all latent representations produced by level $l$ and one-hot-encoded step $t$ as inputs and extracts information that is relevant for the forward model in level $l-1$ at step $t$. For all levels other than the highest level in the hierarchy, the forward models also receive the output of $c$.

**Loss function.** HKSL computes a loss value at each timestep within each level in the hierarchy as the normalized $\ell_2$ distance between a nonlinear projection of the forward model's prediction and the "true" latent representation produced by the level's momentum encoder. Using this "noisy" approximation of the target ensures smooth changes in the target between learning steps and is hypothesized to reduce the possibility of collapsed representations (Grill et al., 2020; Tarvainen & Valpola, 2017). We denote the projection model of level $l$ with $w^l$ and the HKSL loss of level $l$ across the minibatch of trajectories $\tau$ can be written as:

$$\mathcal{L}_{HKSL}^l = \sum_{t=1}^{N} \mathbb{E}_{a,o \sim \tau} \left\| w^l \left( f^l(z_t^l, \bar{a}_t, c^{l+1,l}(\cdot)) \right) - e_m^l(o_{t+n^l}) \right\|_2^2, \tag{1}$$

where $N$ is the number of steps that a given level can take in $\tau$.

**HKSL and SAC.** We make a few adjustments to the base SAC algorithm to help HKSL fit naturally. For one, we replace the usual critic with an ensemble of $h$ critics. Each critic and target critic in the ensemble receive the latent representations produced by a given level's encoder and momentum encoder, respectively. We allow critics' gradients to update their encoders' weights, and each critic is updated using $n$-step returns where $n$ corresponds to the $n$ of the level within which the critic's

given encoder resides. By matching encoders and critics in this way, we ensure encoder weights are updated by gradients produced by targets of the same temporal coarseness.

Second, the actor receives a concatenation of the representations produced by all online encoders. HKSL's actors will make better-informed action selections because they can consider information in the environment that moves at varying levels of temporal coarseness. Finally, we modify the actor's loss function to use a sum of Q-values from all critics:

$$\mathcal{L}_{actor} = -\mathbb{E}_{a \sim \pi, o \sim \tau} \left[ \sum_{l=1}^{h} [Q^l(o,a)] - \alpha \log \pi(a|[e_o^1(o)|...|e_o^h(o)]) \right]. \tag{2}$$

## 5 EXPERIMENTS

We evaluate HKSL with a series of questions and compare it against several relevant baselines. First, is HKSL more sample efficient in terms of agent-environment interactions than other representation learning methods (§ 5.2)? Second, what is the efficacy of each of HKSL's components (§ 5.3)? Third, how well do HKSL's encoders capture task-relevant information relative to our baselines' encoders? (§ 5.4)? Finally, what does does $c^{l,l-1}$ consider when providing information to $l-1$ from $l$ (§ 5.4)?

### 5.1 EXPERIMENTAL SETUP

**Baselines.** We use DrQ (Yarats et al., 2021), CURL (Laskin et al., 2020a), PI-SAC (Lee et al., 2020b), DBC (Zhang et al., 2021) and DreamerV2 Hafner et al. (2021) as our baselines. DrQ regularizes Q-value learning by averaging temporal difference targets across several augmentations of the same images. CURL uses a contrastive loss similar to CPC (van den Oord et al., 2018) to learn image embeddings. PI-SAC uses a Conditional Entropy Bottleneck (Fischer, 2020) auxiliary loss with both a forward and backward model to learn a representation of observations that capture the environment's transition dynamics. DBC uses a bisimulation metric and a probabilisitc forward model to learn representations invariant to task-irrelevant features. DreamerV2 is a model-based method that performs planning in a discrete latent space. All model-free methods use SAC (Haarnoja et al., 2018a;b) as the base RL algorithm, while DreamerV2 leverages an on-policy actor-critic method with a $\lambda$-target critic (Schulman et al., 2016). All methods use the same encoder, critic, and actor architectures to ensure a fair comparison. Additionally, each method uses the same image augmentation. See Appendix C for hyperparameter settings.

**Environments.** We use six continuous-control environments provided by MuJoCo (Todorov et al., 2012) via the DMControl suite (Tassa et al., 2018; 2020), a popular set of environments for testing robotic control algorithms. Each of the six environments uses episodes of length 1k environment steps and a set number of action repeats that controls the number of times the environment is stepped forward with a given action. We use five variations of each DMControl environment for a total of 30 tasks. Four of the variations use distractors provided by the Distracting Control Suite API (Stone et al., 2021), and the fifth variation uses no distractors. We use the "color" and "camera" distractors on both the "easy" and "medium" difficulty settings. The color distractor changes the color of the agent's pixels on each environment step, and the camera distractor moves the camera in 3D space each environment step. The difficulty setting controls the range of color values and the magnitude of camera movement in each task[3].

Additionally, we use an environment of our design, which we call "Falling Pixels". In Falling Pixels, the agent controls a platform at the bottom of the screen and is rewarded +1 for each pixel it catches. Pixels fall from the top of the screen and are randomly assigned a speed when spawned, which controls how far they travel downwards with each environment step. See Appendix B for further information on the environments.

### 5.2 SAMPLE EFFICIENCY

**Training and evaluation procedure.** In our training scheme, agents perform an RL and representation learning gradient update once per action selection. Every 10k environment steps in DMControl

---

[3]Refer to (Stone et al., 2021) for details.

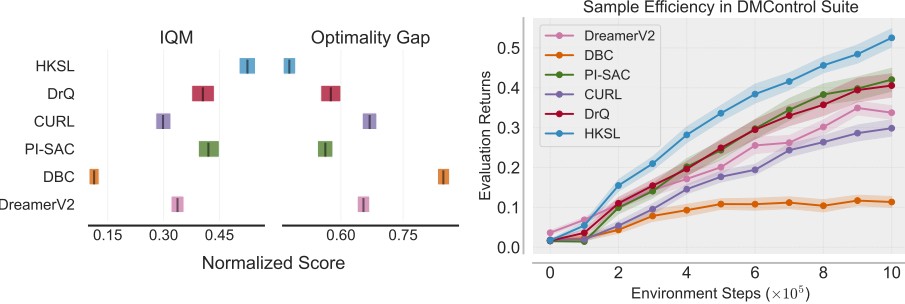

Figure 2: IQM (left) and optimality gap (middle) of evaluation returns at 100k environment steps, and IQM throughout training (right) across all 30 DMControl tasks. Shaded areas are 95% confidence intervals.

and 2.5k environment steps in Falling Pixels, we perform an evaluation checkpoint, wherein the agent's policy is sampled deterministically as the mean of the produced action distribution, and we compute the average performance across 10 episodes. All methods are trained with a batch size of 128. We train agents for 100k and 200k environment steps for five seeds in DMControl and Falling Pixels, respectively.

**Results.** We use the "rliable" package (Agarwal et al., 2021) to plot statistically robust summary metrics in our evaluation suite. To produce aggregate metrics, we normalize all DMControl returns to the maximum per-episode returns, which is 1k for all tasks. Specifically, Figure 2 shows the interquartile mean (IQM) (left) and the optimality gap (middle) along with their 95% confidence intervals (CIs) that are generated via stratified bootstrap sampling[4] at the 100k steps mark in DM-Control. Optimality gap measures the amount by which a given algorithm fails to achieve a perfect score[5]. Additionally, Figure 2 shows IQM and 95% CIs as a function of environment steps (right) in DMControl. Both of these results show that HKSL significantly outperforms our baselines across our 30 environment DMControl testing suite. See Appendix E for individual environment results. We note that simply using a forward model does not guarantee improved performance, as suggested by the comparison between HKSL, PI-SAC, and DBC.

Due to the randomness in Falling Pixels, the maximum per-episode return is difficult to calculate. Therefore, we do not aggregate Falling Pixels with DMControl returns, but instead show the IQM and 95% CIs for Falling Pixels as a function of environment steps in Figure 3 (left). We highlight that HKSL significantly outperforms all of our baselines, converging to a performance of collecting over 20% more pixels per episode than the next-best-performing algorithm. Collecting a large number of pixels in Falling Pixels requires agents to keep track of environment objects that move at varying speeds. HKSL explicitly achieves this with its hierarchy of forward models. Also, we note that DreamerV2 struggles to outperform a random policy. We hypothesize that this is due to Falling Pixels' observation space characteristics: the important information is single-pixel-sized. Hafner et al. (2021) show that image-reconstruction gradients are important to DreamerV2's success (Figure 5 in Hafner et al. (2021)), and the small details in Falling Pixels cause an uninformative reconstruction gradient[6].

## 5.3 COMPONENT ABLATIONS

We probe each component of HKSL to determine its contribution to the overall RL policy learning process. Specifically, we test SAC without the hierarchical model but with HKSL's ensemble of critics (No Repr), HKSL where each level in the hierarchy moves with a single step (All $n = 1$), HKSL without $c$ (No $c$), HKSL where each level in the hierarchy shares encoders (Shared Encoder), and single-level HKSL ($h = 1$). The No Repr ablation tests whether HKSL's performance boost

---

[4]For all plots, we performed at least 5,000 samples.

[5]We note that a perfect score (optimality gap = 0) is technically impossible in the DMControl suite. As such, only the relative positioning of CIs should be considered.

[6]Hafner et al. (2021) also give this reason for why DreamerV2 does poorly in the "Video Pinball" environment.

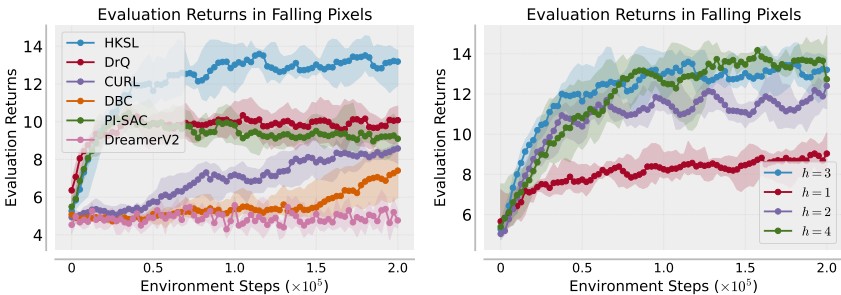

Figure 3: IQM and 95% CIs of evaluation returns for all algorithms in Falling Pixels (left) and ablations over HKSL's $h$ (right).

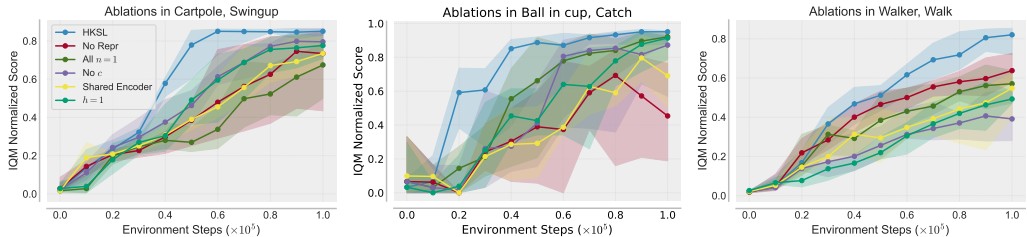

Figure 4: IQM 95% CIs of evaluation returns for HKSL ablations in Cartpole, Swingup (left), Ball in Cup, Catch (middle), and Walker, Walk (right).

is due to the ensemble of critics or the hierarchical model itself. The All $n = 1$ ablation tests our hypothesis that only learning representations at the environment's presented temporal coarseness can miss out on important information. The No $c$ ablation tests the value of sharing information between levels. The Shared Encoder ablation tests if one encoder can learn information at varying temporal coarseness. Finally, the $h = 1$ ablation tests the value of the hierarchy itself by using a standard forward model (e.g., (Schwarzer et al., 2021; McInroe et al., 2021)).

See Figure 4 for the performance comparison between these ablations and full HKSL in the no distractors setting of Cartpole, Swingup, Ball in Cup, Catch, and Walker, Walk. All results are reported as IQMs and 95% CIs over five seeds. We highlight that variations without all components perform worse than full HKSL. This suggests that HKSL requires each of the individual components to achieve its full potential.

Also, we ablate across the number of levels $h$ in HKSL's hierarchy in Falling Pixels. Figure 3 (right) depicts IQMs and 95% CIs over five seeds for values of $h$ in the set $\{1, 2, 3, 4\}$ with temporal coarseness of levels set to $[1, 3, 5, 7]$ for levels one through four, in order. We highlight that increasing $h$ achieves a monotonic improvement in evaluation returns up to when $h = 4$. We hypothesize that setting $h = 3$ captures all relevant information in Falling Pixels, and increasing to $h = 4$ leads to similar returns as when $h = 3$ and does not destabilize learning.

### 5.4 REPRESENTATION ANALYSIS

**How well do representations align with task-relevant information?** To test the ability of encoders to retrieve task-relevant information from pixel input, we save the weights of the encoders for each method throughout training in our evaluation suite. We then use the representations produced by these encoders to train a linear projection (LP) to predict task-relevant information over varying timescales. This process is akin to linear probing (Alain & Bengio, 2017), a method used to analyze representations (e.g,. (Anand et al., 2019)). We note that the encoders' weights are frozen, and the gradient from the prediction task only updates the LP's weights.

In the Cartpole, Swingup task, the objective is to predict the cart's and pole's coordinates. In the Ball in Cup, Catch task, the objective is to predict the ball's coordinates. We collect 10 and five episodes of image-coordinate pairs in each environment for LP training and testing, respectively. We repeat

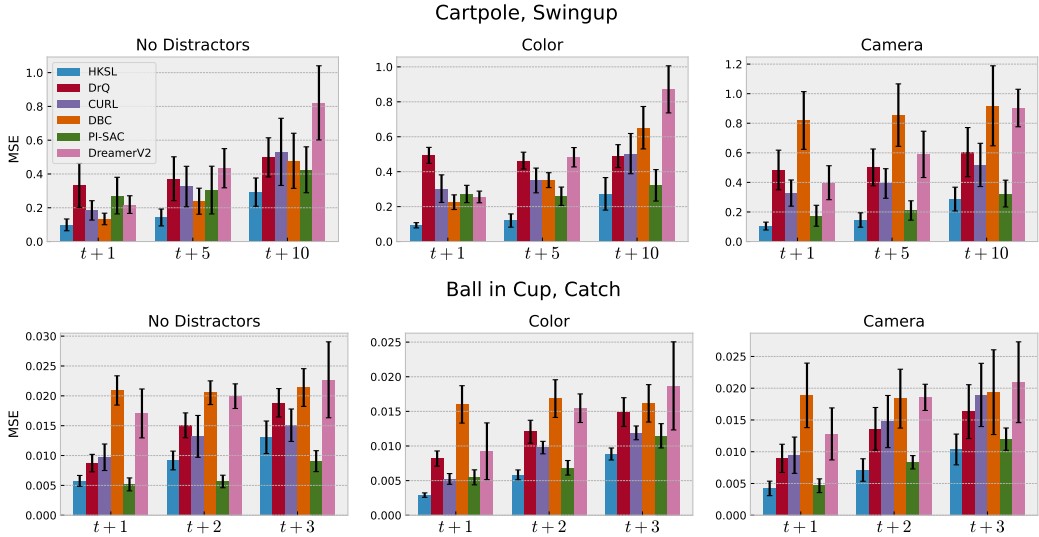

Figure 5: MSE on task-relevant information in unseen episodes for Cartpole, Swingup (top) and Ball in Cup, Catch (bottom) at the 100k environment steps mark. Non-distraction, color distractor, and camera distractor settings shown from left-to-right. Lower is better.

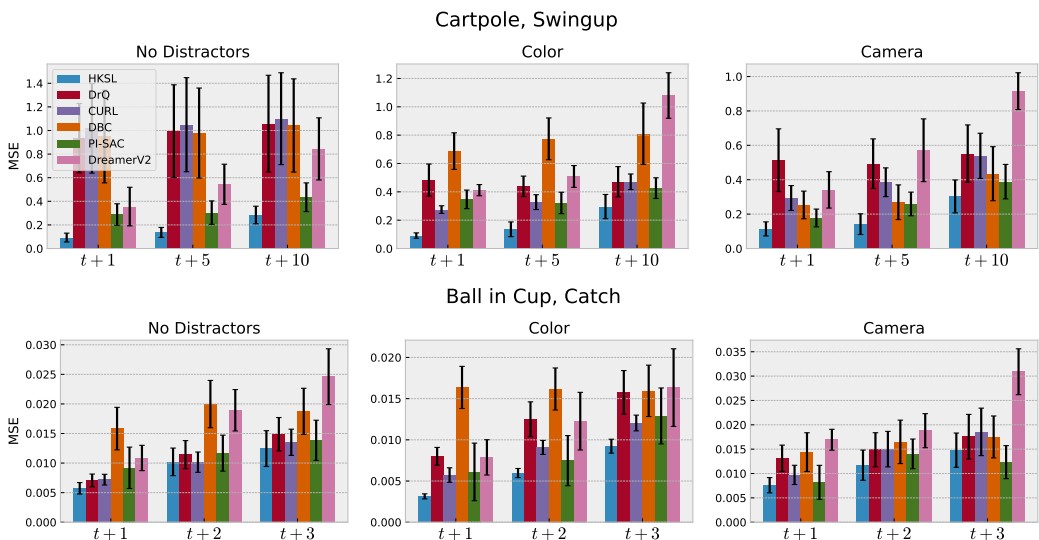

Figure 6: MSE on task-relevant information in unseen episodes for Cartpole, Swingup (top) and Ball in Cup, Catch (bottom) at the 50k environment steps mark. Non-distraction, color distractor, and camera distractor settings shown from left-to-right. Lower is better.

this data-collection exercise for both environments' non-distraction, easy color distractors, and easy camera distractors versions. After fitting the LP on the training sets, we measure the mean squared error (MSE) on the unseen testing set. Figure 5 shows the average MSE and $\pm$ one standard deviation over the testing episodes using encoders trained for 100k environment steps in our benchmark suite. In Cartpole, Swingup (top row), we use the LP to predict coordinates from one ($t + 1$), five ($t + 5$) and 10 ($t + 10$) steps into the future. In Ball in Cup, Catch (bottom row), we use the LP to predict coordinates from one ($t + 1$), two ($t + 2$) and three ($t + 3$) steps into the future. We highlight that HKSL's encoders produce representations that more accurately capture task-relevant information with the lowest variance in nearly every case. Also, this accuracy carries over into the distraction settings, giving a reason for HKSL's relatively strong performance in the presence of distractors, despite not addressing distractors explicitly.

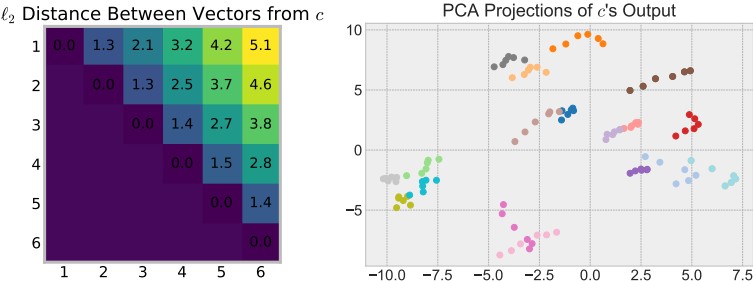

Figure 7: Average distance between vectors produced by $c$ (left). The numbers along the side and bottom correspond to the value of $t$. PCA projections of representations produced by $c$ for multiple timesteps across 18 trajectories (right) with colors corresponding to trajectories.

We repeat this process using encoders from earlier in the agent-training process. Figure 6 shows the MSE and $\pm$ one standard deviation over the testing episodes using encoders trained for 50k environment steps in our benchmark suite. We note that the same pattern from the 100k environment steps encoders persists. These results suggest that HKSL agents benefit from more informative representations in earlier stages of training than our baselines, which leads to better sample efficiency.

**What does $c$ consider?** We hypothesize that the communication manager $c^{l,l-1}$ provides a wide diversity of information for $f^{l-1}$ by taking into account the current transition of the below level $l-1$ as well as the representations from the above level $l$. To check this hypothesis, we perform two tests. First, we measure the $\ell_2$ distance between the vectors produced by $c$ when the step $t$ is changed and other inputs are held fixed. If $c$ completely ignores $t$, the distance between $c(\cdot, 1)$ and $c(\cdot, 4)$, for example, would be zero. Second, we examine the separability of $c$'s outputs on a trajectory-wise basis. If two sampled trajectories are very different, then the representations produced by the above level should change $c$'s output such that either trajectory should be clearly separable.

We first train an HKSL agent where $h = 2$, $n^1 = 1$, and $n^2 = 3$ in Cartpole, Swingup for 100k environment steps and collect 50 episodes of experiences with a random policy. Then, we randomly sample a trajectory from this collection and step through the latent space with both forward models. We repeat this 100 times and measure the pairwise $\ell_2$ distance between $c$'s outputs for every value of $t$ within sampled trajectories. Figure 7 (left) reports the average distance between each pair. We note that the distance between $c$'s output grows as the steps between the pairs grows. This suggests that $c$ considers the transition of the level below it when deciding what information to share. Additionally, we highlight that the distance increases consistently where pairs that are the same number of steps apart are about the same distance apart. For example, pairs $(2, 5)$ and $(3, 6)$ are both three steps apart and share roughly the same average $\ell_2$ distance. This suggests that $c$ produces representations that are grouped smoothly in the latent space. Figure 7 (right) visualizes the PCA projections of $c$'s outputs from 18 randomly sampled trajectories, where each trajectory is a different color. This figure confirms our second intuition, as the representations are clearly separable on a trajectory-wise basis with representations smoothly varying across steps within the same trajectory.

## 6 CONCLUSION

This paper presented Hierarchical $k$-Step Latent (HKSL), an auxiliary task for accelerating control learning from pixels via a hierarchical latent forward model. Our experiments showed that HKSL's representations can substantially improve the performance of downstream RL agents in pixel-based control tasks, both in terms of converged returns and sample efficiency. We also showed that HKSL's representations more accurately capture task-relevant information than our baselines and do so early in training. Finally, we showed that the communication manager organizes information in response to the above and below levels.

## 7 REPRODUCIBILITY STATEMENT

We open-source all code required to train HKSL. The anonymous URL on page two links to this code. Also, we open-source the code for our Falling Pixels environment by placing it within the supplementary .zip file. Finally, we release all data from the training runs for each of the algorithms used in this study, which can be found in the supplementary file.

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

## A  Extended Background

**Soft Actor-Critic.** Soft Actor-Critic (SAC) (Haarnoja et al., 2018a;b) is a popular off-policy, model-free RL algorithm for continuous control. SAC uses a state-action value-function critic $Q$ and target critic $\bar{Q}$, a stochastic actor $\pi$, and a learnable temperature $\alpha$ that weighs between reward and entropy: $\mathbb{E}_{o_t, a_t \sim \pi}[\sum_t \mathcal{R}(o_t, a_t) + \alpha \mathcal{H}(\pi(\cdot|o_t))]$.

SAC's critic is updated with the squared Bellman error over historical trajectories $\tau = (o_t, a_t, r_t, o_{t+1})$ sampled from a replay memory $\mathcal{D}$:

$$\mathcal{L}_{critic} = \mathbb{E}_{\tau \sim \mathcal{D}}[(Q(o_t, a_t) - (r_t + \gamma y))^2], \tag{3}$$

where $y$ is computed by sampling the current policy:

$$y = \mathbb{E}_{a' \sim \pi}[\bar{Q}(o_{t+1}, a') - \alpha \log \pi(a'|o_{t+1})]. \tag{4}$$

The target critic $\bar{Q}$ does not receive gradients, but is updated as an exponential moving average (EMA) of $Q$ (e.g., He et al. (2020)). SAC's actor parameterizes a multivariate Gaussian $\mathcal{N}(\mu, \sigma)$ where $\mu$ is a vector of means and $\sigma$ is the diagonal of the covariance matrix. The actor is updated via minimizing :

$$\mathcal{L}_{actor} = -\mathbb{E}_{a \sim \pi, \tau \sim \mathcal{D}}[Q(o_t, a) - \alpha \log \pi(a|o_t)], \tag{5}$$

and $\alpha$ is learned against a static value.

## B  Environments

Table 1 outlines the action space, the action repeat hyperparameter, and the reward function type of each environment used in this study. The action repeat hyperparameters that are displayed in the table are the standards as defined by (Hafner et al., 2019) and are the same used in most studies in DMControl. The versions of each environment with distractors follow the presented information as well.

The Falling Pixels environment is rendered as a $35 \times 15$ grayscale image. The agent is confined to the bottom row and pixels are spawned at the top row. The agent is placed randomly along the bottom row and the top row is filled with pixels at the beginning of each episode. With each environment step, the pixels travel downwards until they reach the bottom row. If the agent is occupying a pixel's column when it reaches the bottom row, that pixel is "collected" and the agent is rewarded +1. Regardless of whether a pixel is collected, it disappears from the board once it reaches the bottom row. When a column does not have a pixel within it, there is a 2.5% chance for a new pixel to be spawned in that row each environment step. When spawned, the pixel is assigned a speed from the set $\{1, 3, 5\}$ uniformly at random. Each episode is 250 environment steps.

Table 1: Dimensions of action spaces, action repeat values, and reward function type for all six environments in the DMControl benchmark suite and Falling Pixels.

| Environment, Task | $dim(\mathcal{A})$ | Action Repeat | Reward Type |
|---|---|---|---|
| Finger, spin | 2 | 2 | Dense |
| Cartpole, swingup | 1 | 8 | Dense |
| Reacher, easy | 2 | 4 | Sparse |
| Cheetah, run | 6 | 4 | Dense |
| Walker, walk | 6 | 2 | Dense |
| Ball in Cup, catch | 2 | 4 | Sparse |
| Falling Pixels | 1 | 1 | Dense |

Table 2: SAC Hyperparameters used to produce paper's main results.

| Hyperparameter | Value |
|---|---|
| Image padding | 4 pixels |
| Initial steps | 1000 |
| Stacked frames | 3 |
| Evaluation episodes | 10 |
| Optimizer | Adam |
| $(\beta_1, \beta_2)$ Optimizer | (0.9, 0.999) |
| Learning rate | $1e-3$ |
| Batch size | 128 |
| Q function EMA | 0.01 |
| Encoder EMA | 0.05 |
| Target critic update freq | 2 |
| $dim(z)$ | 50 |
| $\gamma$ | 0.99 |
| Initial $\alpha$ | 0.1 |
| Target $\alpha$ | $-|\mathcal{A}|$ |
| Replay memory capacity | 100,000 |
| Actor log stddev bounds | [-10,2] |

## C ARCHITECTURE AND HYPERPARAMETERS

### C.1 SAC SETTINGS

All encoders follow the same architecture as defined by (Yarats et al., 2020). These encoders are made of four convolutional layers separated by ReLU nonlinearities, a linear layer with 50 hidden units, and a final layer norm opertion (Ba et al., 2016). Each convolutional layer has 32 3×3 kernels and the layers have a stride of 2, 1, 1, and 1, respectively. This in contrast to the encoder used in the PI-SAC study (Lee et al., 2020b), which uses Filter Response Normalization (Singh & Krishnan, 2020) layers between each convolution.

The architectures used by the SAC networks follow the same architecture as deinfed by (Yarats et al., 2020). Both the actor and critic networks have two layers with 1024 hidden units, separated by ReLU nonlinearities. This is in contrast to the networks used in the PI-SAC study, which uses a different number of hidden units in the actor and critic networks.

Several studies have shown that even small differences in neural network architecture can cause statistically signficant differences in performance (Islam et al., 2017; Henderson et al., 2018). As such, we avoid using the original PI-SAC encoder and SAC architectures to ensure a fair study between all methods.

Table 2 shows the SAC hyperparameters used by all methods in this study. For method-specific hyperparameters (e.g., auxiliary learning rate, architexture of auxiliary networks, etc.), we defaulted to the settings provided by the original authors.

## C.2 HKSL Hyperparameters

Table 3 shows the hyperparameters that control HKSL. $h$ represents the number of levels, $n$ contains a list of the skips of each level from lowest to highest level, $k$ shows the length of the trajectory sampled at each training step, learning rate corresponds to the learning rate of all HKSL's components, and actor update freq corresponds to the number of steps between each actor update. These hyperparameters were found with a brief search over the non-distractor setting of each environment.

HKSL's communication manager $c$ is a simple two-layer non-linear model. The first layer has 128 hidden units and the second has 50. The two layers are separated by a ReLU nonlinearity.

Table 3: Hyperparameters used for HKSL for each environment.

| Environment, Task | $h$ | $n$ | $k$ | Learning rate | Actor Update Freq |
|---|---|---|---|---|---|
| Finger, spin | 2 | [1,3] | 3 | 1e-4 | 2 |
| Cartpole, swingup | 2 | [1,3] | 6 | 1e-3 | 1 |
| Reacher, easy | 2 | [1,3] | 3 | 1e-4 | 2 |
| Cheetah, run | 2 | [4,5] | 10 | 1e-4 | 2 |
| Walker, walk | 2 | [1,3] | 6 | 1e-3 | 1 |
| Ball in Cup, catch | 2 | [1,3] | 6 | 1e-3 | 1 |
| Falling Pixel | 3 | [1,3,5] | 6 | le-3 | 1 |

## C.3 HKSL's Forward Models

The usual GRU formulation at step $t$:

$$u_{gru}^t = \sigma(f_{gru}^u([a_t|z_{t-1}])) \tag{6}$$

$$r_{gru}^t = \sigma(f_{gru}^r([a_t|z_{t-1}])) \tag{7}$$

$$h_{gru}^t = tanh(f_{gru}^h([r_{gru}^t \odot z_{t-1}|a_t])) \tag{8}$$

$$g_{gru}^t = (1 - u_{gru}^t) \odot z_{t-1} + u_{gru}^t \odot h_{gru}^t \tag{9}$$

where each each distinct $f$ is an affine transform, $\sigma$ is the sigmoid nonlinearity, and $\odot$ is the Hadamard product. In order to allow the forward models to take the optional input from $c$, we add an identical set of additional affine transforms:

$$u_c^t = \sigma(f_c^u([C_t|z_{t-1}])) \tag{10}$$

$$r_c^t = \sigma(f_c^r([C_t|z_{t-1}])) \tag{11}$$

$$h_c^t = tanh(f_c^h([r_c^t \odot C_t|z_{t-1}])) \tag{12}$$

$$g_c^t = (1 - u_c^t) \odot z_{t-1} + u_c^t \odot h_c^t \tag{13}$$

where $C_t$ denotes the output from $c$ at step $t$. Finally, the output of the forward model is the average of the two pathways:

$$z_t = \frac{g_c^t + g_{gru}^t}{2} \tag{14}$$

## D Attention Maps

We examine the encoders within HKSL's hierarchy to ascertain their objects of focus. Each encoder receives gradients relating to a different magnitude of temporal coarseness. Therefore, each encoder should learn to "focus" on different aspects of input images. The top row in each plot shows the unstacked frames that go into the past from right to left (e.g., the framestack depicted with images as $[o_{t-2}, o_{t-1}, o_t]$. The bottom row of each plot shows the attention maps from each encoder. The attention maps are generated by taking the output of the final convolutional layer and averaging across the feature map dimension. All encoders are from HSKL agents after 100k environment steps of training.

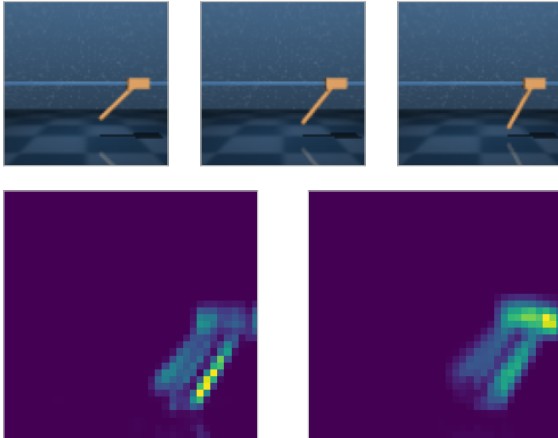

Figure 8: Input frame stack (top row) and corresponding attention maps (bottom row) for a scenario from Cartpole, Swingup. Encoder from first and second level shown on the left and right, respectively.

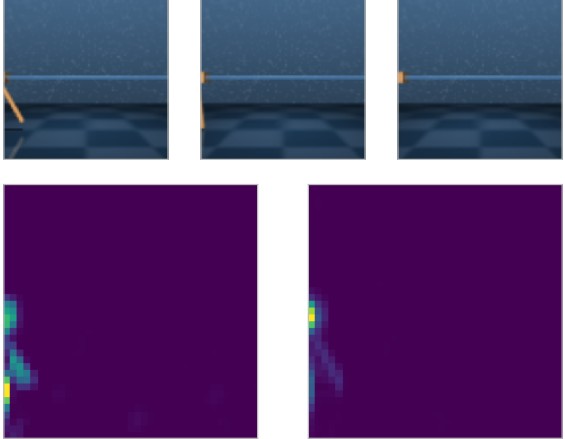

Figure 9: Input frame stack (top row) and corresponding attention maps (bottom row) for a scenario from Cartpole, Swingup. Encoder from first and second level shown on the left and right, respectively.

Figure 8 depicts a scenario from Cartpole, Swingup. We note that the encoder from the first level (left) attends to the pole, an object that is not controlled by the agent. In contrast, the encoder from the second level (right) attends to the cart, which is directly controlled by the agent. Figure 9 also depicts a scenario from the Cartpole, Swingup environment. Here, the cart is offscreen for one frame in the stack. Here, we see the same pattern as in Figure 8. The encoder from the first and second level pay more attention to the pole and the cart, respectively.

Figure 10 depicts a scenario from the Ball in Cup, Catch environment. We highlight that the encoder from the first level (left) appears to attend entirely to the information from the most recent frame in the input stack. In contrast, the encoder from the second level (right) gathers the full trajectory of information from each frame in the stack. This phenomenon is especially apparent in Figure 11,

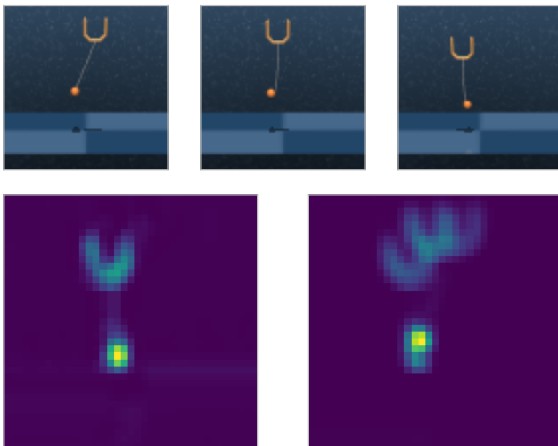

Figure 10: Input frame stack (top row) and corresponding attention maps (bottom row) for a scenario from Ball in Cup, Catch. Encoder from first and second level shown on the left and right, respectively.

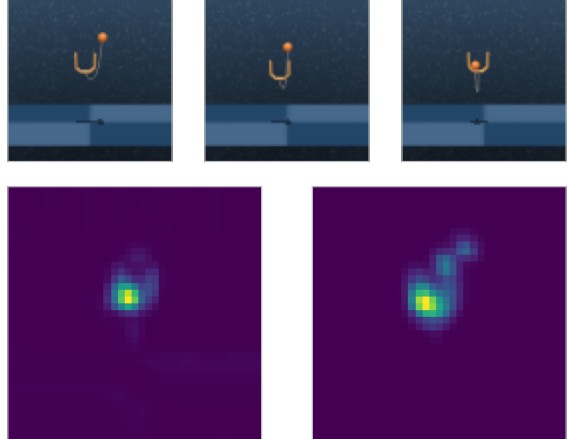

Figure 11: Input frame stack (top row) and corresponding attention maps (bottom row) for a scenario from Ball in Cup, Catch. Encoder from first and second level shown on the left and right, respectively.

where the encoder from the second level (right) captures the trajectory of the ball as it falls into the cup.

# E   INDIVIDUAL ENVIRONMENT RESULTS

This section shows the mean (bold lines) ± one standard deviation (shaded area) for every individual environment and distractor combination. Figure 12 displays the non-distractor environments, Figure 13 shows the color distractors on the easy setting, Figure 14 shows the color distractors on the medium setting, Figure 15 shows the camera distractors on the easy settings, and Figure 16 shows the camera distractors on the medium setting.

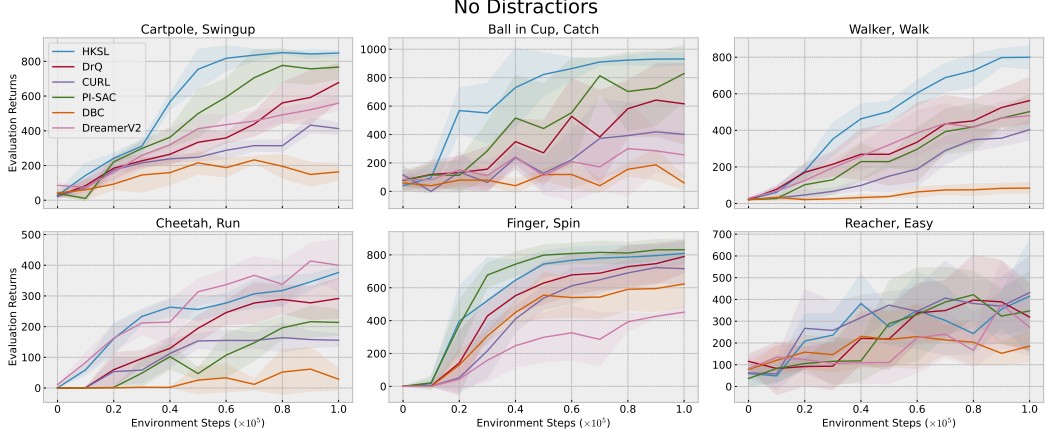

Figure 12: Evaluation returns for agents trained in DMControl without distractors. Bold line depicts the mean and shaded area represents +− one standard deviation across five seeds.

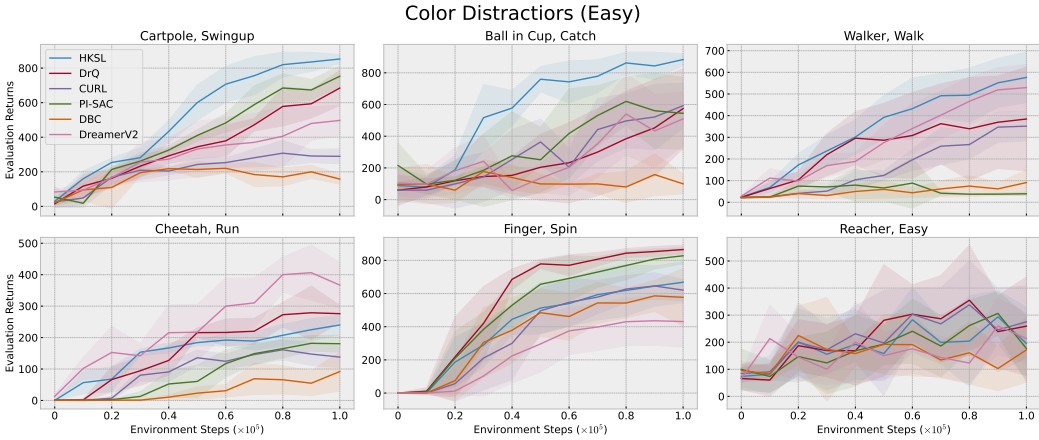

Figure 13: Evaluation returns for agents trained in DMControl with color distractors on the easy setting. Bold line depicts the mean and shaded area represents +− one standard deviation across five seeds.

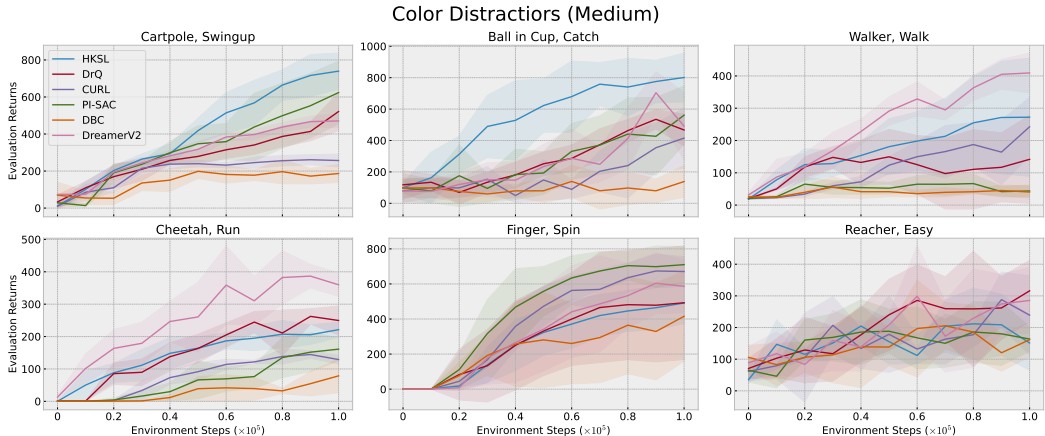

Figure 14: Evaluation returns for agents trained in DMControl with color distractors on the medium setting. Bold line depicts the mean and shaded area represents +− one standard deviation across five seeds.

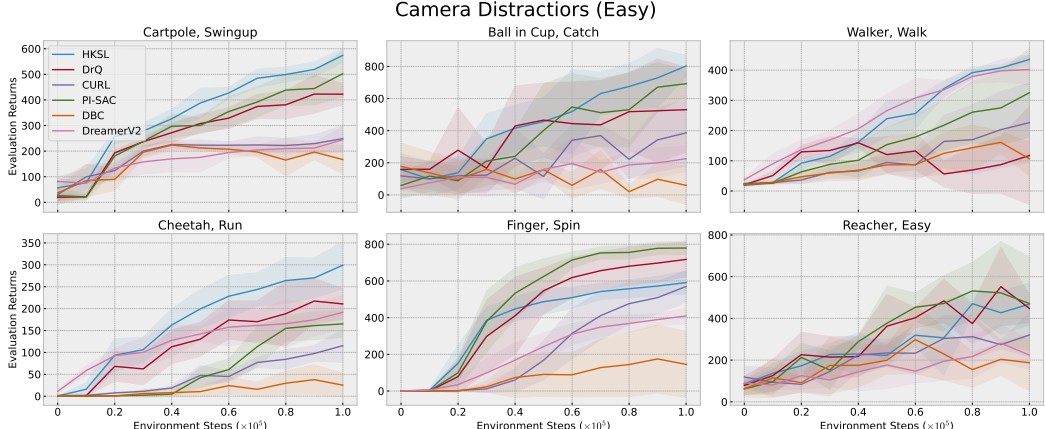

Figure 15: Evaluation returns for agents trained in DMControl with camera distractors on the easy setting. Bold line depicts the mean and shaded area represents +− one standard deviation across five seeds.

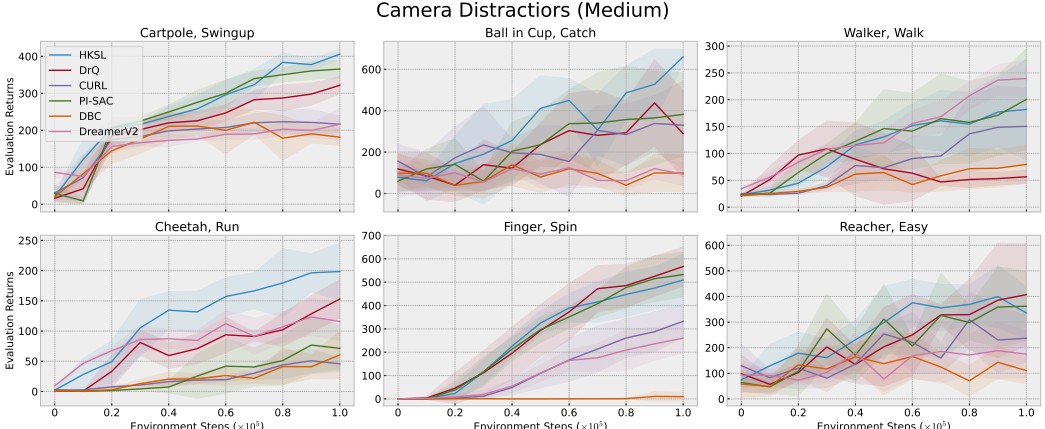

Figure 16: Evaluation returns for agents trained in DMControl with camera distractors on the medium setting. Bold line depicts the mean and shaded area represents +− one standard deviation across five seeds.

