# OpenReview forum: "Learning Representations for Reinforcement Learning with Hierarchical Forward Models"
_ICLR.cc/2023/Conference — Submitted to ICLR 2023_

### Official Review · Reviewer_6JA8 · 2022-10-20

**Confidence:** 4
**Correctness:** 3
**Technical Novelty And Significance:** 2
**Empirical Novelty And Significance:** 2
**Recommendation:** 5

**Clarity, Quality, Novelty And Reproducibility:**

Clarity: The paper is well written and the approach is reasonably well explained. While some of the notation is a bit dense, Fig 1 makes it easier to understand the overall approach. Some of the details of the implementation are hidden in the appendix (Table 3), would be good to explicitly call this out. And some statements (e.g. see point 6 in Weaknesses) don't fall out directly from the results; would be good to explain these better.

Quality & Novelty: The specific use of hierarchical models to learn representations for sample efficient RL is somewhat novel but the results are only marginally better than related work, and some key ablations, asymptotic performance results & baselines need to be included to quantify the contributions better.

Reproducibility: The draft provides a reasonable amount of details and builds on top of existing prior work so in my opinion it can be reproduced with a reasonable amount of effort.

**Details Of Ethics Concerns:**

No concerns

**Strength And Weaknesses:**

Strengths:
1. The paper provides an interesting approach towards training a hierarchy of forward models (albeit at explicitly specified levels of temporal abstraction) and shows that the resulting method performs well on a standard set of benchmarks.
2. The addition of the communication module to communicate information from higher to lower levels is novel and seems to help improve performance on the tested domains.

Weaknesses:
1. The key novelty of the approach is on the addition of a hierarchy of forward models but there is no clear ablation in the paper on the sensitivity of the approach to the choice of hyper parameters of the hierarchy. Specifically there is no ablation across the number of levels used (h), and the number of steps skipped by a given level (n^l). The only relevant ablation provided was with [h=1] or [all n=1]. The appendix mentions that these parameters were chosen after a short search -- would be great if all the results across this search are presented. This would provide more empirical evidence to the stability and generality of the provided approach which is currently hard to evaluate.
2. Table 3 of the appendix presents hyperparameters used with HKSL. There are 5 different sets of parameters for the 7 tasks presented in the paper -- parameters varied include the trajectory length (k), number of levels (h), number of skipped steps (n^l) and learning rate. Most prior work (e.g. DrQ, CURL) uses atmost one or two sets of hypers for a variety of tasks. Is there a particular reason to tune the hypers for HKSL? Relating to the previous question, is it due to the sensitivity of HKSL to hypers? Would be good to clarify this.
3. On the topic of hypers, how does "k" affect the number of sequential predictions a model needs to make? For "Finger Spin" and "Reacher Easy" k = 3 but n_l = [1, 3]. Does this mean that the higher level forward model (n_1 = 3) does only a single forward prediction? Please clarify this in the text.
4. All the results presented are only for 100k env steps. Most prior work reports results on 100k and 500k env steps as well as asymptotic performance; is there a reason why this is not done in the proposed approach? It would be useful to add these results as it would be good to know what the asymptotic performance of the method is, not just whether the data efficiency in the low sample complexity regime is good.
5. All the presented baselines use either some form of representation loss or single-step prediction losses. Is there a reason to not compare to a baseline that uses multi-step prediction losses, or some form of reconstruction objective (e.g. Dreamer)? While reconstruction approaches can potentially fail on the distracting control suite they might do quite well on some of the analysis results presented (such as MSE prediction error of object positions). Would be a useful comparison to add.
6. Re: Fig 7, it was mentioned that the data used for this comparison was collected using random policies. Why is this data interesting to consider for this particular analysis? Wouldn't it make sense to look at trajectories that are temporally consistent in some form for this comparison? It is also mentioned in the analysis that "This suggests that c considers the transition of the level below it when deciding what information to share", how do the results from Fig 7 (left) show this? From my perspective it only shows that the error scales proportionally to "t", not that the communicator (c) uses knowledge from the lower-level (which it doesn't have access to) to decide which information to communicate. It would be great if this point is explained better.
7. Out of the tasks tested in the paper only the "Falling Pixels" task introduced in the paper explicitly requires some form of reasoning across temporal scales. It would be helpful if the approach was tested on atleast one other task (e.g. from Atari?) where temporal abstraction of some form was necessary.

**Summary Of The Paper:**

This paper presents an approach towards data-efficient reinforcement learning that leverages a hierarchy of latent forward models to train representations at different levels of temporal abstraction. The models are designed to take in encoded observations at a given timestep t and a sequence of actions (where sequence length >= 1) and can predict future encodings of observations at different timescales depending on the coarseness of the corresponding model. Additionally there is a communication network that passes information from forward models at slower temporal scales to more fine-grained temporal scales. The models are trained using squared error between predictions and "true" latents, and are effectively used to shape the learning of observation representations; the representations from different levels are fed into the actor to predict the final action output by the policy. The approach (combined with SAC) is tested on several continuous control tasks from the DMControl suite along with variations such as distractors, and changes to camera pose or object colors; an additional task of "falling pixels" that potentially requires reasoning at different temporal abstractions is also used as a test environment. Comparisons are made against baselines which use different objectives to shape representations on top of SAC. Overall it performs better than the compared baselines at low sample complexity (100k transitions) thereby being more data-efficient. A few ablations and analysis of the learned representations are provided.

**Summary Of The Review:**

Overall, while the approach presents a novel application of hierarchies of forward models to shape representations for data-efficient RL, the results are a bit weak. Some ablations, asymptotic performance results and analysis of the sensitivities of the approach are missing. As such I would recommend to reject this paper in its current form.

---

> ### Author Response · Authors · 2022-11-16
> **Response to reviewer 6JA8**
>
> We thank reviewer 6JA8 for their comments and questions. We are glad reviewer 6JA8 found our approach to be interesting and our paper to be written clearly. Please see below for comments on your questions. Also, see the revised paper, where changes are highlighted in blue. We would be grateful if reviewer 6JA8 could reconsider their score based on our clarifications and revisions, or let us know if further clarifications are required.
>
> **Ablations over levels in hierarchy and coarseness**
>
> Please see our answer in the general post titled "Ablations over levels in hierarchy and coarseness".
>
> **Why different hyperparameters per task?**
>
> HKSL was designed to capture environment information that moves at varying levels of temporal coarseness. This characteristic may differ slightly between environments.
>
> **Relationship between $k$ and a level's temporal coarseness.**
>
> Your interpretation is correct. We state this in the paper directly below Eqn 1: ``where $N$ is the number of steps that a given level can take in $\tau$". For example, if $k=9$ and the temporal coarseness of levels are $[1,3]$, the top level would make three predictions in total and the bottom level would make nine.
>
> **Reason for only examining the low-data regime instead of the asymptotic limit**
>
> Please refer to our answer in the general post titled "Measuring asymptotic performance".
>
> **Additional baselines (e.g., Dreamer)**
>
> In the revised paper, we have included results from a strong model-based baseline, DreamerV2 (Hafner et al. 2021). Please refer to our answer in the general post titled "Comparison to model-based methods" for more information.
>
> **Why random policies for Fig 7?**
>
> We note that deploying a deterministic policy (regardless of its training extent) would create a dataset of trajectories with low diversity. Even if the initial states between episodes are different, deterministic policies in DMControl environments would quickly cause trajectories to coalesce. If portions from two trajectories are identical, then we cannot expect a deterministic transformation (e.g., a forward pass through $c$) to separate the two. Instead, using a random policy produces a dataset of trajectories with high diversity.
>
> **Questions about $c$ and Figure 7**
>
> We would like to clear up any misunderstandings about the communication manager $c$ and the data displayed in Figure 7.
>
> $c$ has access to information from both the level above it and the level below it. $c$ receives the representations produced from the forward models from the level above. Also, $c$ has access to the one-hot-encoded timestep $t$, which represents the current transition number that the below-level is on. Having access to information from both sides of the communication process allows $c$ to extract relevant information from the above level's rollout as it pertains to the below level's step.
>
> Figure 7 (left) does not display an error metric. Instead, it displays the $\ell_2$ distance between representations. This figure shows that representations produced by $c$ change smoothly through a trajectory.
>
> Figure 7 (right) shows how $c$ leverages information from above and below levels when determining what information to output. The trajectories as a whole are clearly separable, which is influenced by the input from the above level. When the trajectory being fed through HKSL changes, $c$ produces an output vector that is on a different portion of the representation manifold, hence the separability by trajectory. Also, the steps within the trajectories are clearly separable, which is influenced by the value of $t$, which is information from the below level. When we feed the same trajectory through HKSL but change $t$, $c$ produces a vector that is near others within the same trajectory, but not the exact same vector. This property is confirmed by the data displayed in Figure 7 (left).

---

> > ### Comment · Reviewer_6JA8 · 2022-11-19
> > **Response to rebuttal**
> >
> > Thanks for addressing some of my comments with the rebuttal, it is good to see ablations across some of the hypers used by the approach and a comparison to Dreamer which adds more evidence on the strengths of the proposed approach. One question re using random trajectories for Fig. 7, it makes sense that a random policy would provide diversity but the actual long-horizon behavior of the resulting trajectories might be meaningless and hence I feel that any analysis on this behavior does not add much value or insight. Is it possible instead to use random trajectories but with temporally correlated noise -- this would still give diversity while also ensuring that the trajectories are temporally coherent. Lastly, my concerns regarding the number of hyper-parameters and variations for each task were not really addressed which I think is quite important for reproducibility and practical use of the proposed approach. Overall, I will increase my score to a weak reject in light of the rebuttal.

---

### Official Review · Reviewer_EsdY · 2022-10-21

**Confidence:** 5
**Correctness:** 4
**Technical Novelty And Significance:** 2
**Empirical Novelty And Significance:** 3
**Recommendation:** 6

**Clarity, Quality, Novelty And Reproducibility:**

Quality
--------
The quality of the paper is good. The approach is sound, the claims are substantiated and the evaluation is sufficiently thorough and considers reasonable baselines.
However, I think that the chosen coarseness levels have not been sufficiently well discussed and evaluated. For the DMC control task the paper only considers two levels. Only for the falling pixel environment more than two levels (3) have been evaluated. Given that the topic of the paper is a hierarchical architecture, it would be crucial to evaluate the effects of the number of chosen layers, and the chosen coarseness levels.


Clarity
--------
The paper is well written, and mostly clear. However, I have a few questions that I would like the authors to address:
1. What is the purpose of the projection layer? If we omit it in Eq. 1, the l2 loss would try to match the actual latent with the predicted latent, which is reasonable, although I see the problem of mode collapse (an encoder that always predicts the same loss would achieve result in zero loss). How does the projection layer alleviate this problem?
2. For the plots in Fig. 5 and 6: Which encoder was used for computing the HKSL MSE? Did you concatenate the different embeddings (as in the policy input)? I would expect to use the corresponding embedding depending on how many steps ahead we want to predict.
3. There seem to be only 18 different colors in Fig. 7 (right), although there should be 20 trajectories. Why?

Originality
-------------
The proposed architecture is novel and interesting. Although there are no theoretical justification for the specific choices, the architecture is sounds and its evaluation thereby interesting.

Reproducibility
--------------------
The paper seems to be well reproducible, as the code is published and in a good state.
The procedure for tuning the temporal coarseness and the number of hierarchies should be discussed.

**Strength And Weaknesses:**

- The paper tackles an important problem on how to capture and predict the effects of the agent's actions at different temporal coarseness levels. (+)
- The method/architecture is sound and achieves good performance (+)
- The paper is well written (+)
- The results seem to be well reproducible, code published (+)
- A few parts are a bit unclear (-)
- The paper does not require theoretical analysis or derivations, so the novelty is limited to the network architecture (-)

**Summary Of The Paper:**

The main contribution of the paper is a hierarchical architecture for predicting observations at different levels of temporal coarseness, and its evaluation for RL from images.

The individual dynamic models for the different temporal coarseness levels are independent from each other and each consist of an encoder, a latent-dynamics model and a nonlinear projection network. Furthermore, except for the lowest level dynamic model, each model also contains a communication manager that summarizes all latent predictions within a trajectory segment and provides it to the latent dynamic model of the next lower level as additional input.

The models are evaluated using a modified SAC agent, which uses an example of Q function: one critic per temporal coarseness level. Furthermore, the policy gets all embeddings (1 per coarseness level) of the current observation as input. The dynamic model itself, however, is only used for representation learning, but not directly used during RL. The modified SAC agent is evaluated at the DMControl suite and on a new toy task "falling pixels". The result show that the proposed method can outperform suitable baselines in terms of sample efficiency.


**Summary Of The Review:**

The paper is well written and of high quality, with sufficient novelty.
However, I outlined a few issues that I would like the authors to address:
1. Ablations of critical hyperparameters (number of hierarchies, chosen coarseness) are missing.
2. There are some issues regarding the clarity

---

> ### Author Response · Authors · 2022-11-16
> **Response to reviewer EsdY**
>
> We thank reviewer EsdY for their feedback. We are glad reviewer EsdY found that our paper is well written and that our method is sound. Please see below for comments on your questions. Also, see the revised paper, where changes are highlighted in blue. We would be grateful if reviewer EsdY could reconsider their score based on our clarifications and revisions, or let us know if further clarifications are required.
>
> **Additional analysis over hierarchy levels/coarseness**
>
> Please see our answer in the general post titled "Ablations over levels in hierarchy and coarseness".
>
> **What is the purpose of the projection layer $w$?**
>
> The work of Chen et al. (2020) establishes empirically that a non-linear projection can improve the representation quality of the preceding layer. (See section 4.2 entitled "A nonlinear projection head improves the representation quality of the layer before it".) Additionally, the projection layer, in combination with a slow moving-average target encoder, makes HKSL's prediction pipeline similar to BYOL (Grill et al. 2020). Grill et al. (2020) find empirically that this prediction objective prevents representation collapse (See Table 5b in Grill et al. (2020)). As we state in the "Loss function" subsection of Section 4 in our paper, this prediction pipeline produces a "noisy" approximation of the true target, which is hypothesized to prevent representation collapse (Tarvainen \& Valpola 2017).
>
> **For the plots in Fig. 5 and 6: Which encoder was used for computing the HKSL MSE? Did you concatenate the different embeddings (as in the policy input)? I would expect to use the corresponding embedding depending on how many steps ahead we want to predict.**
>
> The representations for all of HKSL's encoders were concatenated together. We studied this formulation because HKSL's actor receives this concatenation. Also, we have added a qualitative analysis to probe what each encoder in HKSL's hierarchy is attending to in Appendix D. For more information, please refer to our answer in the general post titled "What do the encoders of each level attend to?"
>
> **There seem to be only 18 different colors in Fig. 7 (right), although there should be 20 trajectories. Why?**
>
> Thank you for highlighting this discrepancy. It appears we miscounted, as the plot only contains 18 trajectories, and therefore only 18 different colors. This typo has been corrected in the revised paper.

---

### Official Review · Reviewer_X4PZ · 2022-10-22

**Confidence:** 3
**Correctness:** 3
**Technical Novelty And Significance:** 3
**Empirical Novelty And Significance:** Not applicable
**Recommendation:** 6

**Clarity, Quality, Novelty And Reproducibility:**

See my comments above. One other small fix for the authors:
* “?? shows the MSE and ± one standard deviation over the testing episodes using encoders” – ?? should be Figure 6 I think


**Strength And Weaknesses:**

Strengths
* Clarity of writing and presentation. The paper is very easy to read. The motivation is clearly laid out. The method is clearly explained for either a casual reader (main text) or more detailed reader (mathematically laid out in the appendix). The results are sign-posted and follow sensible questions to ask given the setup of the approach.
* Statistics and reproducibility. The authors use interquartile means to show that their approach statistically outperforms recent baselines, as well as the various ablations of their model. The comparisons seem sound, and the diversity of experiments underscores the general utility of the hierarchical model-based representation learning technique. The authors also provide a link to code to ensure that results are reproducible. I commend the authors for their scientific openness and careful scholarship.
* Novelty and impact. The idea of using hierarchical models for reinforcement learning is not novel for model-based reinforcement learning, where the hierarchical models are used for planning. However, I am not aware of work that uses hierarchical models in model-free reinforcement learning as a way of learning better representations that can ignore distractors. I thought this was a nice twist, and this paper would invite more investigation from others looking at hierarchical models for model-based RL to think about using the same approaches for generic representation learning in model-free RL instead.

Weaknesses
* Overall I really enjoyed reading this paper. However, I had one big question related to the framing of the method as being useful for sample efficiency. The presented approach is model-free, and results are presented after a *tiny fraction* of the number of environment steps often shown for model-free algorithms. The paper highlights the usefulness of their approach (HKSL) for exactly this kind of efficiency. BUT normally we would use model-based techniques to improve sample efficiency, which are absent from this paper. It would be particularly compelling if the presented method can outperform model-based techniques when environment samples are limited, or if the presented method still outperforms model-free baselines even in the limit of significant experience.
  * To address this concern, could the authors provide results for one of two experiments:
    * Compare to a model-based reinforcement learning method that uses the learned hierarchical models for planning, or really any other model-based RL technique, for the low-sample regime? I would like to understand whether the improvements here in model-free actually rival what one would expect for model-based.
    * Compare results for massive amounts of experience as well (ie show the asymptotic behavior of all the presented baselines, not just their behavior after 50k or 100k steps). If HKSL, the presented method, still outperforms baselines in the limit of significant experience, that would ameliorate my concern about “why not just use model-based RL”.




**Summary Of The Paper:**

This paper demonstrates how learning an ensemble of hierarchical models that predict latent representations at varying step-sizes into the future can be used for representation learning in model-free reinforcement learning. The proposed hierarchical technique is not used for planning, but rather the representations learned using their technique enable more efficient model-free reinforcement learning from pixels than baseline techniques. The paper provides compelling ablation results, baseline comparisons across 30 environments plus one newly introduced pixel-based control task, and releases all code needed to train their model and run the new environment.

**Summary Of The Review:**

Overall I think this paper is clear, novel, potentially impactful, and represents sound and good science. I am recommending acceptance, but would reduce my score if the authors do not respond to my questions about model-based vs. model-free RL techniques.

---

> ### Author Response · Authors · 2022-11-16
> **Response to reviewer X4PZ**
>
> We thank reviewer X4PZ for their comments. We are glad reviewer X4PZ found our paper to be clear and our method to be well-motivated. Please see below for comments on your questions. Also, see the revised paper, where changes are highlighted in blue. We would be grateful if reviewer X4PZ could reconsider their score based on our clarifications and revisions, or let us know if further clarifications are required.
>
> **Comparison to model-based methods**
>
> In the revised paper, we have included results from a strong model-based baseline, DreamerV2 (Hafner et al. 2021). Please refer to our answer in the general post titled "Comparison to model-based methods" for more information.
>
> **Compare results for massive amounts of experience to show asymptotic performance**
>
> Please refer to our answer in the general post titled "Measuring asymptotic performance".
>
> **“?? shows the MSE and ± one standard deviation over the testing episodes using encoders” – ?? should be Figure 6 I think**
>
> Thank you for pointing this out. The figure reference has been corrected in the updated version of the paper.

---

### Official Review · Reviewer_WeT7 · 2022-10-24

**Confidence:** 4
**Correctness:** 3
**Technical Novelty And Significance:** 2
**Empirical Novelty And Significance:** 2
**Recommendation:** 6

**Clarity, Quality, Novelty And Reproducibility:**

The paper is relatively clear in terms of what it is doing, though the method is somewhat unintuitive. The writing does not detract from the paper. While methods doing multi-step dynamics modeling have been used in RL, having multiple encoders appears to be a novel contribution. The experiments are somewhat weak, since the domains are such that the number of layers remains low, and it is not clear what the layers are learning that is distinct between each other.

**Strength And Weaknesses:**

The reasoning that single step forward models fail to capture relevant information is a bit weak in the context of an MDP, where the markov property should ensure that all information is captured in a single state.

It isn't clear exactly how the encoder is prevented from collapse, since if the encoder is encoded to zero state then the forward models would have zero loss.

While the results are promising and the writing is clear, the intuition behind what the different levels of the hierarchy might be encodinig. In particular, it would be nice ot have some visualization demonstrating how the higher levels capture different information from the lower ones, and which parts of the encoding the policy attends to. This might be more informative than indicating if task-relevant components are attended to, since it isn't clear why hskl would be especially effective at capturing task-relevant components. Otherwise, it is possible that a multi-head representation might get similar results, without the heads having temporally different meanings.

The experiments have many domains with only two levels, except for the falling pixels. It might be nice to look at a very long horizon tsk with a large number of levels.


**Summary Of The Paper:**

Learn a hierarchy of latent models where each level takes in the latent state and a length n concatenation of actions and predicts the state after n steps, instead of simply predicting single step transitions. Information is passed from the layers predicting longer temporal distance to the lower levels. The representations are trained with l2 loss between the representations and the encoded observations. The representation is then used by SAC with h critics, corresponding to different levels of the hierarchy, and a policy representation that is based on the encodings learned by the levels.

**Summary Of The Review:**

I think this paper can be accepted because it appears to provide a novel, simple change to existing model-based state representation RL methods, and it is implemented well enough with sufficient experiments to constitute a contribution.

---

> ### Author Response · Authors · 2022-11-16
> **Response to reviewer WeT7**
>
> We thank reviewer WeT7 for their feedback. We are glad reviewer WeT7 found our paper to be clear and to have sufficient support for our claims. Please see below for comments on your questions. Also, please see the revised paper, where changes are highlighted in blue. We would be grateful if reviewer WeT7 could reconsider their score based on our clarifications and revisions, or let us know if further clarifications are required.
>
> **The reasoning that single step forward models fail to capture relevant information is a bit weak in the context of an MDP, where the markov property should ensure that all information is captured in a single state.**
>
> Please note that we use partially-observable MDPs (POMDPs). The agent observes pixel images which typically do not give full information about the underlying state of the process, and thus the Markov property may not hold from the perspective of the agent (Igl et al. 2018, Jaakkola et al. 1994).
>
> Due to the lack of the Markov property, multi-step forward models are useful because they explicitly try to capture information over many time steps. In contrast, single-step models may tie together information over the time-axis inefficiently. Our results show that this is the case in many environments.
>
> **It isn't clear exactly how the encoder is prevented from collapse**
>
> The projection layer $w$, in combination with a slow moving-average target encoder, makes HKSL's prediction pipeline similar to BYOL (Grill et al. 2020). Grill et al. (2020) find empirically that this prediction objective prevents representation collapse (See Table 5b in Grill et al. (2020)). As we state in the "Loss function" subsection of section 4 in our paper, this prediction pipeline produces a "noisy" approximation of the true target, which is hypothesized to prevent representation collapse (Tarvainen \& Valpola 2017).
>
> **Difference in information-capture between hierarchy levels**
>
> Please refer to our answer in the general post titled "What do the encoders of each level attend to?"
>
> **Multi-head representation**
>
> By multi-head representation, do you mean an encoder with one head per level? If so, our "Shared Encoder" ablation in section 5.3 addresses this. In terms of the representation learning objective, sharing a single encoder between all levels is a multi-head representation, as each level will have a unique "head" in the form of the projector $w$. The Shared Encoder ablation shows that it is difficult for a single encoder to produce good representations across varying temporal coarsenesses. These ablation results suggest that encoders of different hierarchies encode different information.
>
> **Additional tasks with long horizons**
>
> Our study focuses on the setting where we learn behavior with limited environment interactions; therefore, we did not include long-horizon tasks. However, we agree that adding tasks that require agents to reason over long horizons would be an interesting future direction to study.

---

### Author Response · Authors · 2022-11-16
**General Response**

We thank the reviewers and chair for their time and useful feedback. Some reviewers asked similar questions, so we are posting our responses to those questions here. Also, please see the revised paper version, where changes are highlighted in blue.

**Measuring asymptotic performance**

We highlight that the results in Falling Pixels displays several algorithms at convergence. Here, HKSL converges to a performance superior to our baselines' converged performance.

Also, we emphasize that the research question we study in our work can be phrased as: "given a small number of agent-environment interactions, how can we learn a well-performing control policy with RL?" We motivate this question in our introduction. This research question is important to focus on because real-world RL deployments cannot reasonably collect millions of agent-environment interactions. As such, we aim to develop algorithms that can learn a well-performing control policy in only a small number of agent-environment interactions.

**Comparison to model-based methods**

Please see the revised paper version. In it, we included training runs and analysis for DreamerV2 (Hafner et al. 2021). We highlight this addition with blue text in Section 5.1.

We note that DreamerV2 is a strong baseline but does not outperform HKSL. We show this for Falling Pixels in Figure 3, and the DMControl suite in Figure 2. We have also added DreamerV2 to our representation analysis in Figure 5 and Figure 6. We highlight that the representations produced by DreamerV2's encoders do not capture information across time as well as HKSL's encoders.

**Ablations over levels in hierarchy and coarseness**

The ablations shown in section 5.3 provide a view for only one level ($h=1$) and when coarseness is matched between levels (All $n=1$) in the DMControl suite. We have also added an ablation in Falling Pixels in section 5.3 and Figure 3 (right), where we test the impact on the hyperparameter $h$ (number of levels in HKSL's hierarchy). We note that we observe a monotonic increase in evaluation returns as $h$ increases, up to a point (when $h=4$). We hypothesize that $h=3$ captures all relevant information in Falling Pixels, and therefore increasing $h$ beyond three keeps results stable but at the same level as three.

**What do the encoders of each level attend to?**

We have added a qualitative analysis in Appendix D titled ``Attention Maps" that provides insight into the differences in information-capture between hierarchy levels. We highlight that the encoders do appear to attend to different pieces of information in the environment. For example, the figures suggest that, in Cartpole, Swingup, the encoder from level one focuses on the movement of the pole, while the encoder from level two focuses on the cart.

---

### Author Response · Authors · 2022-12-09
**Review follow up**

Dear Reviewers and Area Chair,
We thank you for your reviews and feedback.

We are glad that the reviewers found our work interesting and our method intuitive. We hope that our updated paper and responses have addressed the raised concerns. As the discussion period is ending, please let us know if we still need to address any concerns or if you have anything else you would like to discuss.

Thank you for your time,

Paper Authors

---

### Decision · Program_Chairs · 2023-01-20

**Decision:**

Reject

**Justification For Why Not Higher Score:**

Overall, starting with the motivation, following through the design choices and the task choices, there is something not entirely adequate about how the design choices are motivated.  Due to the many moving parts that potentially required tuning, I'm not convinced that without the clarity of motivation, the approach is likely to be sufficiently impactful to justify acceptance.

**Justification For Why Not Lower Score:**

N/A

**Metareview: Summary, Strengths And Weaknesses:**

The authors introduce a hierarchical k-step auxiliary task for representation learning to capture structure at multiple temporal resolutions to accelerate model-free RL from pixel-based inputs.  As motivation, they note that forward models are one kind of auxiliary representation learning objective that could increase data efficiency, but observe that 1-step forward models may be less relevant than temporally abstract forward models, if the tasks are long-horizon.

The reviews for this paper were ultimately all borderline, with 3/4 reviewers leaning weakly positive and the remaining reviewer leaning weakly negative.  More specifically, reviewers tended to find the paper clear though not necessarily very well motivated, and with some arbitrary design choices having been made (variously described as unintuitive, not theoretically motivated, etc.).  Reviewer WeT7 notes that it is odd that the task didn't really require long horizons, despite that being part of the motivation.  Reviewer X4PZ requested a model-based approach as a baseline.  The authors included DreamerV2 as a baseline.  Reviewer 6JA8 originally identified lack of ablations of number of hierarchy layers as an omission.  The authors added this ablation and that, along with other clarifications and the added baseline, prompted the reviewer to update their score (3-->5).


**Summary Of Ac-Reviewer Meeting:**

Because this paper was borderline, we had a video chat to discuss the paper.  Unfortunately reviewer X4PZ was unable to participate.  Reviewer 6JA8 explained their original score and why they updated it, indicating that the fact that the tasks don't feature a need for hierarchy/abstraction (with the exception of the falling pixels task) was strange.  Reviewer EsdY emphasized that theoretical novelty is limited, and the paper is mostly empirical. However, the paper is well presented, sound, and well carried out.  The problem is interesting and there is some benefit to the innovation. Reviewer WeT7 noted that hierarchical actor-critic / RL has been around, but model-free uses of hierarchical forward models are possibly new.  However, WeT7 argued that hierarchy is potentially more elegant over actions because timescale is multiplicative, whereas with multiple resolutions of forward models you don't get this (i.e. longer timescales aren't multiplicatively longer).  In addition,  experiments aren't compelling and reiterated that falling pixels is the only task that requires temporal abstraction.  Reviewer EsdY voiced disagreement with the claim that the tasks aren't relevant.

The discussion did help refine my own perspective on the work.  The strongest element of the paper is simply that it performs well (in aggregate, fig 2; though individual environment performance is quite variable, appendix E).  A point that arose in the discussion was that hierarchical forward models as used here might just be generic auxiliary tasks rather than being specifically useful from a temporal abstraction standpoint.  On the other hand, if this effort is specifically about hierarchy and temporal abstraction, this is not very compelling given the choice of tasks.